# Deep-ultraviolet electroluminescence and photocurrent generation in graphene/hBN/ graphene heterostructures

Su-Beom Song[1,2,6], Sangho Yoon[1,2,6], So Young Kim[1,2,3], Sera Yang[1,2], Seung-Young Seo[1,2], Soonyoung Cha[1,2], Hyeon-Woo Jeong[3], Kenji Watanabe [4], Takashi Taniguchi [5], Gil-Ho Lee [3], Jun Sung Kim [2,3], Moon-Ho Jo [1,2] & Jonghwan Kim[1,2,3✉]

Hexagonal boron nitride (hBN) is a van der Waals semiconductor with a wide bandgap of ~ 5.96 eV. Despite the indirect bandgap characteristics of hBN, charge carriers excited by high energy electrons or photons efficiently emit luminescence at deep-ultraviolet (DUV) frequencies via strong electron-phonon interaction, suggesting potential DUV light emitting device applications. However, electroluminescence from hBN has not been demonstrated at DUV frequencies so far. In this study, we report DUV electroluminescence and photocurrent generation in graphene/hBN/graphene heterostructures at room temperature. Tunneling carrier injection from graphene electrodes into the band edges of hBN enables prominent electroluminescence at DUV frequencies. On the other hand, under DUV laser illumination and external bias voltage, graphene electrodes efficiently collect photo-excited carriers in hBN, which generates high photocurrent. Laser excitation micro-spectroscopy shows that the radiative recombination and photocarrier excitation processes in the heterostructures mainly originate from the pristine structure and the stacking faults in hBN. Our work provides a pathway toward efficient DUV light emitting and detection devices based on hBN.

[1] Department of Materials Science and Engineering, Pohang University of Science and Technology, Pohang, Republic of Korea. [2] Center for Artificial Low Dimensional Electronic Systems, Institute for Basic Science (IBS), Pohang, Republic of Korea. [3] Department of Physics, Pohang University of Science and Technology, Pohang, Republic of Korea. [4] Research Center for Functional Materials, National Institute for Materials Science, Tsukuba, Ibaraki, Japan. [5] International Center for Materials Nanoarchitectonics, National Institute for Materials Science, Tsukuba, Ibaraki, Japan. [6] These authors contributed equally: Su-Beom Song, Sangho Yoon. ✉email: jonghwankim@postech.ac.kr

Van der Waals (vdW) semiconductors have emerged as a unique platform for optoelectronic applications[1–7]. The strong light-matter interaction in the atomically thin structures leads to the efficient light emission or absorption properties in the broad energy range from infrared to visible frequencies depending on the bandgap size as shown for transition metal dichalcogenides[1] and black phosphorus[2]. Vertical integration with other vdW materials further enables the injection or collection of charge carriers via the atomically thin insulators of the tunneling barriers and conductors, which provides highly efficient and tunable device architectures for light-emitting diodes[3], photovoltaics[4,5], photosensors[6], and electro-optic modulators[7].

Hexagonal boron nitride (hBN) is an indirect and wide bandgap vdW semiconductor. The conduction band minimum and valence band maximum are located at the M point and the neighborhood of the K point in the momentum space, respectively, forming an indirect gap of ~5.96 eV[8,9]. Despite its indirect bandgap characteristics, hBN exhibits strong photoluminescence[8,10] and cathodoluminescence[11–14] at deep-ultraviolet (DUV) frequencies, which are identified as phonon-assisted radiative recombination from indirect excitons by a recent two-photon excitation photoluminescence study[8] and also by following theoretical studies[12,15,16]. Due to the efficient electron-phonon coupling, the emission lineshape exhibits an unconventional temperature dependence of the linewidth[17,18]. Furthermore, a subsequent study[12] measured the remarkably high internal quantum efficiency of spontaneous light emission, ~50 % at 10 K, which is even comparable to that of direct bandgap semiconductors. The intriguing optical properties of hBN as a wide bandgap semiconductor can lead to optoelectronic devices at DUV and higher frequencies for important applications[19,20] ranging from the photocatalysis and the sterilization to the chemical analysis and the short-distance communication. In contrast to $Al_{1-x}Ga_xN$ emissive layers, which has been most intensively studied for DUV light-emitting devices[19], photon extraction from hBN emissive layers can be fundamentally more efficient since the emission direction is oriented along the crystalline c-axis[9]. Although a DUV cathodoluminescence device of hBN has been demonstrated by combining with a portable high-voltage (>1 kV) field emitter[21], the previous studies on electroluminescence have been limited to light emission from the deep-level trap states[22,23] or inelastic tunneling processes[24] at near-ultraviolet, visible and near-infrared frequencies in hBN. On the other hand, the photocurrent generation process remains largely elusive in terms of carrier excitation near the band edge[25–31], which can provide important information for understanding the electronic structure and optical properties of hBN.

In this study, we demonstrate DUV electroluminescence and photocurrent generation in vdW heterostructures where a DUV photoactive layer of hBN is vertically stacked between a pair of graphene electrodes. The luminescence and photocurrent measurement at DUV frequencies are carried out by utilizing our femtosecond laser micro-spectroscopy set-up with widely tunable photoexcitation energy across the hBN bandgap. The electric field from the bias voltage efficiently drives photocarriers excited over the hBN bandgap with remarkably high responsivity of over 32 mA/W, corresponding to an external quantum efficiency (EQE) of 20%. On the other hand, strong electric field from high bias voltages enables carrier injection from the electrodes to the hBN band edges by the Fowler-Nordheim tunneling mechanism, which leads to the radiation of prominent DUV electroluminescence lines. Our DUV laser excitation spectroscopy of photoluminescence and photocurrent shows that radiative recombination and photocarrier excitation processes in the vdW heterostructures mainly arise from the pristine structure and the stacking faults of hBN. In conjunction with the recent progress on high quality and wafer-scale growth of hBN[32–35], our study reveals the possibilities of scalable and highly integrated vdW heterostructure devices of hBN for not only efficient electrically-driven applications of light emission and modulation but also sensitive and fast photodetection at DUV and higher frequencies.

## Results

**DUV optoelectronic processes in hBN vdW heterostructures.** The photocurrent generation and electroluminescence processes in hBN vdW heterostructures are schematically illustrated in Fig. 1a, b. The active hBN layer for DUV absorption and emission is stacked between a pair of graphene electrodes (labeled as Gr)

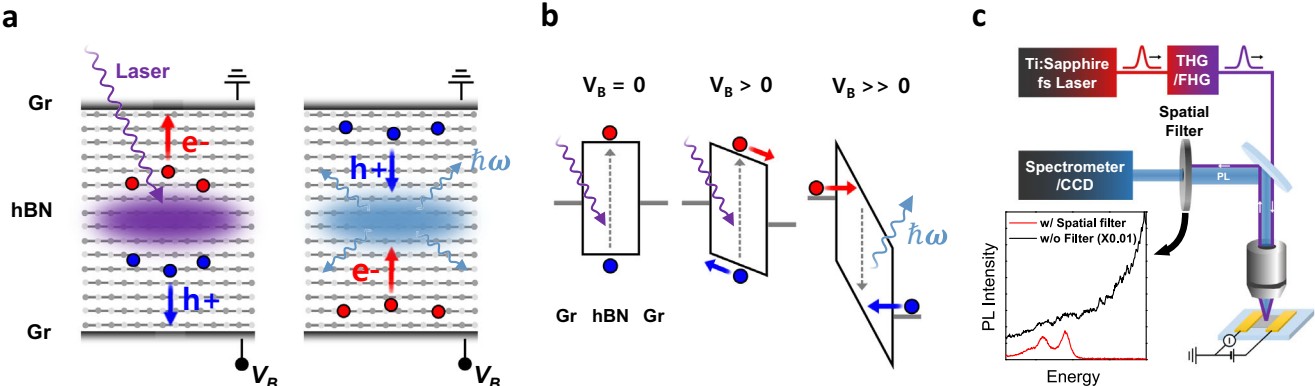

**Fig. 1 DUV optoelectronic processes and femtosecond laser micro-spectroscopy of hBN in vdW heterostructures. a** Schematics of the photocurrent generation and electroluminescence processes in the graphene (Gr)/hexagonal boron nitride (hBN)/graphene (Gr) heterostructure devices. $V_B$, e−, h+, and $\hbar\omega$ represent bias voltage, electrons, holes, Planck's constant and photon energy, respectively. **b** Energy band diagram of the heterostructure devices under $V_B$. Fermi level of graphene electrodes (gray solid lines) is nearly in the middle of hBN bandgap. Photo-excited carriers over the bandgap are driven by the electric field for $V_B > 0$. Charge carriers are tunnel-injected to the band edge of hBN by a high electric field for $V_B \gg 0$, which leads to radiative recombination at DUV frequencies. Gray dashed arrows pointing up and down represent the carrier excitation and radiative recombination, respectively. **c** Illustration of DUV micro-spectroscopy set-up for luminescence and photocurrent measurement with widely tunable laser excitation via fourth and third harmonic generation (FHG and THG) of femtosecond laser pulses. The iris spatially blocks the excitation laser residue before the spectrometer and charge-coupled device (CCD) with the minimal loss of photoluminescence signal (PL). The inset illustrates PL spectra measured with a spatial filter (red solid line) and without a spatial filter (black solid line).

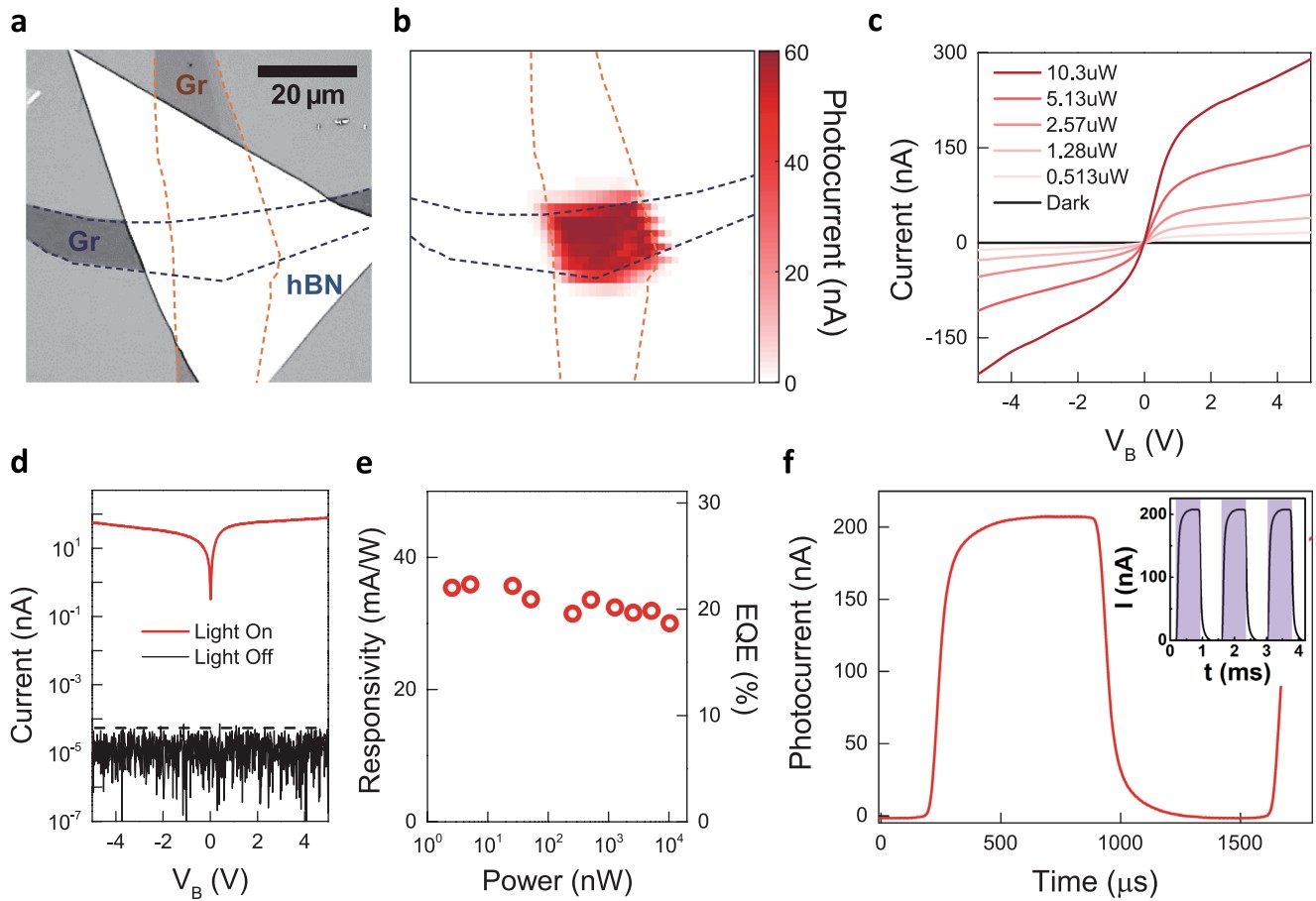

**Fig. 2 Efficient photocurrent generation under laser excitation at 6.22 eV at room temperature. a** Optical microscope image of the representative device for photocurrent measurement. Orange and blue dashed lines represent to the top and bottom graphene electrodes, respectively. **b** Scanning photocurrent image measured at $V_B = 3$ V. A high photocurrent is observed where the hBN overlaps with the graphene electrodes. **c** I-V characteristics in the dark and under laser excitation. **d** I-V characteristics in log scale. The current in the dark condition (Light Off) is dominated by the instrument noise level which is below 50 fA (black dashed line), whereas the device shows high current for the laser power of 2.5 μW (Light On). **e** Laser power dependence of responsivity and photocurrent external quantum efficiency (EQE). **f** Temporal response of the photocurrent. The inset shows the response for the square-wave laser intensity profile (purple-shaded stripes).

that is nearly transparent at the DUV frequency range[36] of our interest. Figure 1b presents the energy band diagram upon the application of the bias voltage ($V_B$). Fermi levels of the graphene electrodes (gray solid lines) are located nearly in the middle of the hBN bandgap[37]. At $V_B = 0$, the conduction and valence bands of hBN are flat, assuming no static charge transfer at the hBN and graphene electrodes interface. Photocarriers relax via radiative and non-radiative recombination processes without generating a net photocurrent. At $V_B > 0$, the electric field creates a moderate potential gradient in the conduction and valence bands, which transports photocarriers to the graphene electrodes. At $V_B \gg 0$, the strong electric field forms a dramatic triangular potential profile through which electrons and holes can be injected to the hBN conduction and valence bands, respectively, from the graphene electrodes by Fowler-Nordheim tunneling mechanism[38–41]. The electrically injected electrons and holes recombine radiatively emitting photons near the bandgap. Under reverse bias, charge carriers are collected or injected from opposite directions, which can also enable photocurrent generation and electroluminescence due to the symmetric device structure. The representative devices for photocurrent and electroluminescence measurements are shown in Figs. 2a and 3a, respectively. Orange and blue dashed lines correspond to the top and the bottom graphene electrodes, respectively. For

electroluminescence measurement, the vdW heterostructure of Gr/hBN/Gr is encapsulated with additional hBN layers to improve the electroluminescence process by providing atomically thin and clean interfaces and protection from the external environment (Supplementary Fig. 1 for electroluminescence spectra from the devices without hBN encapsulation). For photocurrent measurement, the heterostructure device is not encapsulated with hBN because the encapsulation layer can substantially attenuate the DUV laser excitation above the bandgap and suppress photocarrier generation in the active hBN layer. On the other hand, the encapsulation layer does not absorb electroluminescence since the signal emerges below the bandgap. The thicknesses of the active hBN layers for the photocurrent device (Fig. 2a) and the electroluminescence device (Fig. 3a) are 40 and 35 nm, respectively. The detailed device fabrication procedure is provided in Methods.

Figure 1c illustrates our home-built microscopy set-up, where all the optical components are compatible at DUV frequencies, for measuring photocurrent and luminescence spectra. The photoexcitation energy can be widely tuned from 5.89 to 6.3 eV and from 3.54 to 4.65 eV via fourth and third harmonic generation, respectively, of femtosecond laser pulses from the Ti:Sapphire oscillator. The diameter of the focused laser beam spot can be ~2 um using a reflective objective, while the piezo-

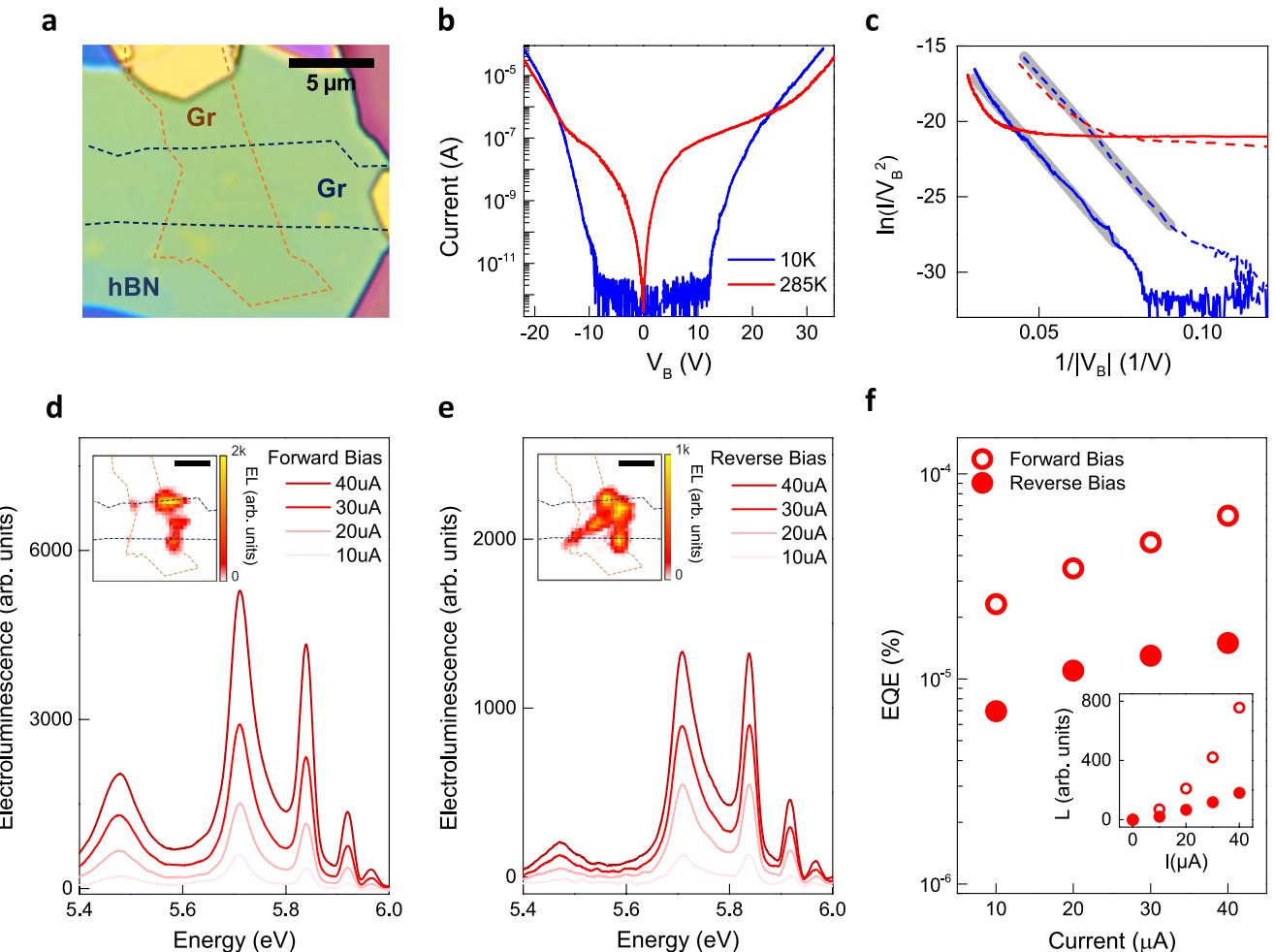

**Fig. 3 DUV Electroluminescence from tunnel-injected charge carriers at room temperature. a** Optical microscope image of the representative device for electroluminescence (EL) measurement (scale bar 5 μm). Top and bottom graphene electrodes (Gr) are marked with orange and blue dashed lines, respectively. **b** I-V characteristics at a temperature of 10 K (blue solid line) and 285 K (red solid line). **c** I-V characteristics replotted with respect to $\ln(I/V_B^2)$ and $1/|V_B|$ under forward bias (solid lines) and reverse bias (dashed lines). Gray solid lines show the linear fitting to our data. **d, e** DUV EL spectra for current from 10 to 40 μA under forward and reverse biases, respectively. The spatial profiles of DUV EL are shown in the insets (scale bar 5 μm). **f** Current dependence of EL EQE. The inset shows a plot of the total-integrated luminescence intensity (L) from 5.4 to 6 eV versus current. Red empty and filled circles correspond to forward and reverse biases, respectively.

stage allows the scanning of the laser spot over the devices. The choice of bandpass or notch filters for eliminating the excitation laser residue is limited at DUV frequencies for the photoluminescence detection. To address this problem, we purposely illuminate the laser beam (purple solid line) at an oblique angle with respect to the luminescence signal collection direction (blue-shaded stripe), as shown in Fig. 1c. The iris spatially blocks the excitation laser residue before the spectrometer and charge-coupled device (CCD) with minimal loss of luminescence signal, which is illustrated in the inset of Fig. 1c. Using our set-up, we can measure not only photocurrent but also photoluminescence spectra via resonant laser excitation continuously across the bandgap while the laser background is significantly suppressed, which has been challenging in the previous studies[8,26,42].

**DUV photocurrent measurements**. The DUV photocurrent measurement shows that the hBN vdW heterostructures can efficiently generate a photocurrent at room temperature for photoexcitation energy above the bandgap of ~5.96 eV. For the excitation energy of 6.22 eV, laser power of 2.5 μW, and bias voltage of 3 V, we observe a high photocurrent signal from the region

overlapped with graphene electrodes by scanning the laser beam over the device (Fig. 2b). For the excitation energy of 4.48 eV and the same laser power and bias, the photocurrent drastically decreases by nearly two orders of magnitude (Supplementary Fig. 3 for the photocurrent mapping at the excitation energy of 4.48 eV) indicating that the generation of high photocurrent is closely relevant to the interband transition in hBN. We will discuss the excitation energy dependence of the photocurrent later, and detailed spectroscopic characterization is provided in Fig. 4f.

The I-V characteristics of the photocurrent (Fig. 2c) indicate that the electric field from the bias voltage drives the photocarrier transport. In the dark condition (black solid line in Fig. 2c, d), no current is observed beyond the noise level from −5 V to 5 V. The nearly absent dark current ($I_{dark}$) results from the large bandgap of hBN and large Schottky barrier heights for both electrons and holes across the hBN and graphene electrodes interface. Moreover, a 40-nm-thick hBN with high crystal quality minimizes the direct or defect-assisted tunneling process[39,43] below the bandgap across the electrodes. Under the photoexcitation with increasing laser power, the device exhibits high and nonlinear current following the polarity and magnitude of the bias voltage. The photocurrent rapidly increases at low voltages and moderately saturates at the

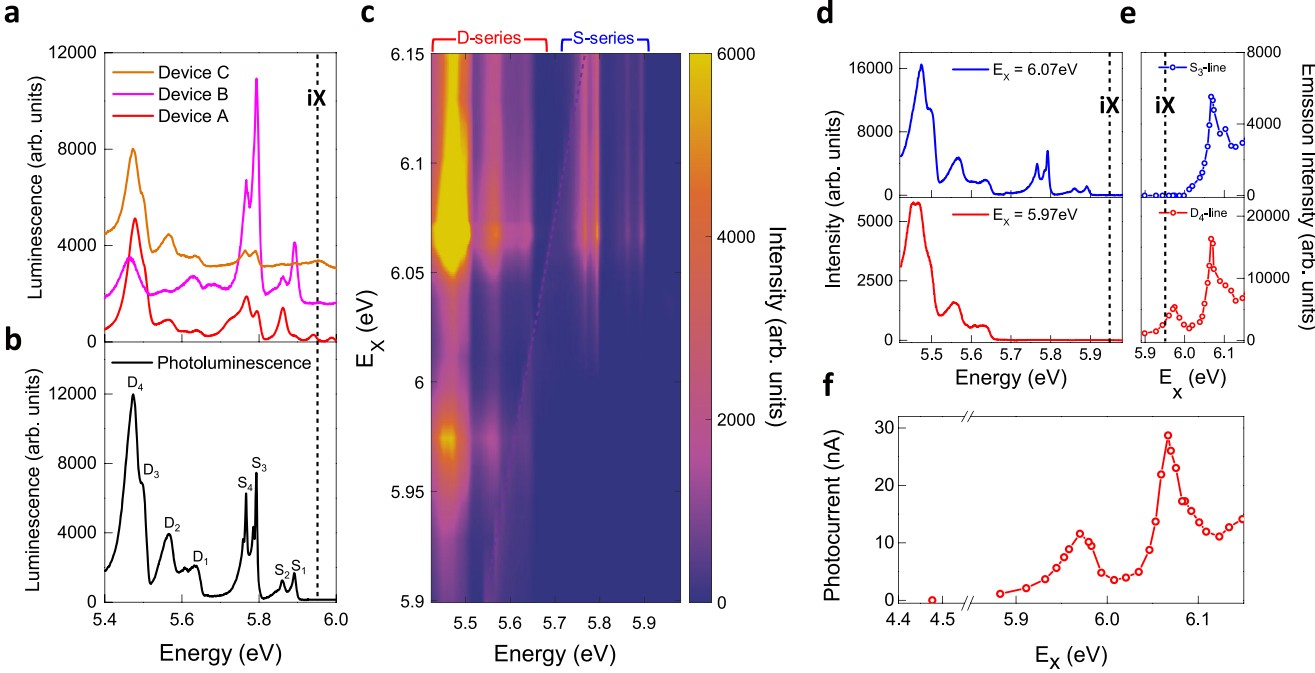

**Fig. 4 DUV laser excitation spectroscopy of photoluminescence and photocurrent at a temperature of 10 K. a** Electroluminescence spectra from the Device A, B and C at 10 K. **b** The representative photoluminescence spectrum from as-exfoliated hBN crystals at 10 K. In the photoluminescence spectrum, narrow lines of intrinsic phonon-assisted emission (peaks at 5.89, 5.86, 5.79 and 5.76 eV) from the indirect gap (iX with black dashed line) are labeled as $S_{1-4}$ lines whereas peaks at 5.63, 5.56, and 5.47 eV, and a shoulder at 5.5 eV in relatively broad spectral profile are labeled as $D_{1,2,4}$ and $D_3$ lines, respectively. **c** 2D color map of the photoluminescence excitation spectra. The vertical axis, horizontal axis and color scale correspond to the photoexcitation energy ($E_X$), energy and intensity of luminescence signal, respectively. Luminescence lines including $S_{1-4}$ and $D_{1-4}$ are grouped into S- and D-series. **d** Horizontal linecuts of (**c**) for $E_X = 6.07$ eV and $E_X = 5.97$ eV. **e** Vertical linecuts of (**c**) for $S_3$-line and $D_4$-line. iX with black dashed line in (**d**, **e**) is labeled for the indirect gap of the pristine structure in hBN. **f** Photocurrent spectrum far below and across the bandgap of hBN with the bias voltage of 3 V.

high voltages. This nonlinear I-V characteristic is commonly observed in metal-semiconductor-metal (MSM) photodetectors with two Schottky contacts back-to-back across the semiconductor channel[20]. Photoconductive detectors with ohmic contacts exhibit linear I-V characteristics because carriers can be readily drawn out from the contacts upon the application of the bias voltage. For MSM detectors, the carrier injection into the semiconductor from the Schottky contacts is not sufficient. Thus, the photocurrent shows moderate saturation behavior as the bias electric field depletes the photo-excited carriers.

We find that the generation of photocurrent is remarkably efficient in the hBN vdW heterostructure. Figure 2e shows the photocurrent at $V_B = 5$ V with varying excitation power from 2 nW to 10 μW. The photocurrent exhibits nearly constant responsivity of ~32 mA/W without any significant power-dependent saturation behavior in the wide dynamic range of the excitation power. The photon-to-current conversion efficiency can be further evaluated with photocurrent EQE expressed as $hcI/eP\lambda$ where $I$, $P$, $\lambda$, $h$, $c$, and $e$ represent the photocurrent, incident power, excitation wavelength, Planck constant, speed of light in vacuum, and electron charge, respectively[4,5]. We find that the EQE is ~20%, which can further increase with the bias voltage. For the temporal response of the photocurrent, we examine the rise and fall times of the photocurrent upon the excitation intensity profile of the square wave using a mechanical chopper. The inset of Fig. 2f shows the typical response for the multiple excitation cycles marked with the purple-shaded region. The detailed analysis of a single cycle (red solid line in Fig. 2f) yields both 10–90% rise and 90–10% fall times of ~50 μs, which is mainly limited by our current amplifier response of ~20 kHz.

This efficient generation and the high-speed response of photocurrent indicates excellent carrier excitation and transport

properties of the photoactive hBN layer in the vdW heterostructures at DUV frequencies. The thickness of few tens of nanometers is sufficient for hBN to absorb significant portion of excitation light[44]. The vdW integration of graphene electrodes enables the efficient and rapid collection of photocarriers from the short channel of hBN with high crystalline quality without significant degradation of the heterointerface. According to a recent study[30], the rise time of a vertical hBN heterostructure can be shorter than 0.7 μs, which stimulates further investigation in the future. Note that our observation about the hBN vdW heterostructures is consistent with the previous studies on other vdW semiconductors, including transition metal dichalcogenides[4,5] and black phosphorus[6] at visible and infrared frequencies.

**DUV electroluminescence measurements.** To investigate the electroluminescence process, charge carriers from graphene electrodes are injected into the band edges of hBN via Fowler-Nordheim tunneling mechanism as described in Fig. 1b. The I-V characteristics show drastic current increases under both forward and reverse bias voltages from ~10 V at a temperature of 10 K (blue solid line in Fig. 3b). The I-V characteristics can be analyzed by using the model based on Fowler-Nordheim tunneling mechanism[38–41]. The Fowler-Nordheim tunneling current can be described by the following equation for a single injection electrode:

$$I = \frac{A_{eff}q^3mV^2}{8\pi h\Phi_B m^* d^2}\exp\left[-\frac{8\pi\sqrt{2m^*}\Phi_B^{3/2}d}{3hqV}\right] \quad (1)$$

V and d represent the bias voltage and insulator thickness, respectively. $A_{eff}$ is the effective tunneling area, $q$ is the elementary charge, $h$ is the Planck's constant, $m$ is the free

electron mass, $m^*$ is the effective mass of the electrons or holes in the insulator, and $\Phi_B$ is the tunneling barrier height for electrons or holes at the electrode and the insulator interface. The equation can be further formulated as

$$\ln\left(\frac{I}{V^2}\right) = \ln\left(\frac{A_{eff}q^3m}{8\pi h\Phi_B m^* d^2}\right) - \frac{8\pi\sqrt{2m^*}\Phi_B^{3/2}d}{3hq}\frac{1}{V} \quad (2)$$

where $\ln(I/V^2)$ and $1/V$ hold distinct linear relationship. The I-V characteristics are replotted with respect to $\ln(I/V_B^2)$ and $1/|V_B|$ separately for forward bias (blue solid line) and reverse (blue dashed line) biases in Fig. 3c. The excellent agreement with linear fitting (gray solid lines) indicates that Fowler-Nordheim tunneling is the major carrier conduction mechanism for both bias polarities. The nearly identical slopes of both the biases indicate that the tunneling current is dominated by a single type of charge carrier with the same effective mass and tunneling barrier height. According to recent studies[40,41], the injection of holes is more efficient than electrons at the graphene and hBN junction via Fowler-Nordheim tunneling, suggesting that holes are the major charge carriers in our tunneling current. The offset between the blue solid line and blue dashed line in Fig. 3c is possibly due to the formation of charged traps[45] and different effective tunneling areas of the anodes (i.e., the electrodes for injecting holes) under the forward and reverse bias voltages. Further experimental studies with the control of work function[40,41] could characterize the electron contribution to the tunneling current as well as the exact tunneling barrier heights, which is beyond the scope of this work. The device exhibits similar I-V characteristics at 77 K (Supplementary Fig. 4), which is expected because Fowler-Nordheim tunneling current does not depend on temperature. In contrast, a significant amount of current flows at room temperature under bias voltages less than 10 V after multiple measurement cycles under high electric fields (red solid line in Fig. 3b). The plot of the I-V characteristics with respect to $\ln(I/V_B^2)$ and $1/|V_B|$ (red solid and dashed lines in Fig. 3c for the forward and the reverse biases, respectively) shows substantially different characteristics compared to that of the tunneling process observed at low temperature. We find that high electric fields can create defects in hBN crystals, which enables temperature-sensitive conducting channels such as Poole-Frenkel emission[46] (Supplementary Fig. 5 for the observation on defect formation). Nevertheless, even at room temperature, the I-V characteristics for I > ~1 μA exhibits that the Fowler-Nordheim tunneling dominates the current in the device and provides electrons and holes for the electroluminescence process.

DUV electroluminescence measurements show that electrically injected electrons and holes radiatively recombine in hBN at room temperature. The spatial profiles of DUV electroluminescence are imaged using a CCD detector with a bandpass filter with a full-width-half-maximum of ~1 eV centered at 5.8 eV. The insets of Fig. 3d, e display the spatial profiles under the forward and reverse biases, respectively. The electroluminescence signal is mostly concentrated in or near the overlapping area of the two graphene electrodes. The signal appears particularly strong at the corners or edges of the area, and similar behavior is observed for other devices (Supplementary Fig. 6 for the spatial profiles of more devices). This can be due to the enhancement of the local electric field at the electrode boundaries, which facilitates carrier injection via Fowler-Nordheim tunneling and horizontally spreads charge carriers outside the overlap area[47]. Figure 3d, e show the electroluminescence spectra for current from 10 to 40 μA under forward and reverse biases, respectively, where multiple luminescence lines emerge from 5.4 to 6 eV. The total-integrated luminescence intensity (L) monotonically increases with the current (inset of Fig. 3f). The electroluminescence EQE (Fig. 3f)

reaches $\sim6.3\times10^{-5}\%$ under forward bias (see Supplementary Note 5 for the measurement of electroluminescence EQE). EQE gradually increases with the current, which is frequently observed in light-emitting devices with low EQE where the radiative recombination process provides a minor contribution to the total current[48,49]. The electroluminescence EQE and spatial profiles show notable discrepancies under the two bias polarities. Considering the I-V characteristics (Fig. 3b) and the inhomogeneous spatial profiles (insets in Fig. 3d, e, Supplementary Figs. 6 and 7), the discrepancy originates from asymmetric charge carrier injection and transport in hBN under high electric fields. See Supplementary Note 6 for more detailed discussion.

The low-temperature measurements of electroluminescence and photoluminescence spectra demonstrate that both electrons and holes are injected to the band edges of hBN. The red solid line in Fig. 4a (labeled as Device A) shows the electroluminescence spectrum at a temperature of 10 K, obtained from the same device as in Fig. 3. Compared to the spectra at room temperature, additional sharp spectral features are observed as the width of luminescence lines becomes narrower at lower temperatures. Notably, other devices (labeled as Device B and C) exhibit qualitatively similar features in electroluminescence spectra. The black solid line in Fig. 4b shows the representative photoluminescence spectrum commonly observed for the as-exfoliated hBN crystals with a laser excitation energy of 6.22 eV. Photoexcited carriers via the band-to-band transition exhibit a series of emission lines that is also present in the electroluminescence spectra with good agreement. The spectral features of electroluminescence, which are not present in the photoluminescence spectrum, are discussed later. According to the previous studies[8,9,12,15,16], the four narrow lines, referred to as the S-series, at 5.89 ($S_1$-line), 5.86 ($S_2$-line), 5.79 ($S_3$-line), and 5.76 eV ($S_4$-line) originate from the intrinsic indirect exciton at ~5.96 eV (labeled as iX marked with black dashed line) in the pristine structure of hBN crystals via the emission of TA, LA, TO, and LO phonons at the T point, respectively, in the momentum space which compensates the momentum mismatch for the conversion of the indirect exciton to the photon in the free space. On the other hand, the relatively broad lines below 5.7 eV, referred as the D-series, including the peaks at 5.63 ($D_1$-line), 5.56 ($D_2$-line), 5.47 eV ($D_4$-line) and the shoulder feature at 5.5 eV ($D_3$-line) are dominantly inherent from defects in the hBN crystals. The physical origin of $D_4$-line, the most prominent feature in the D-series, has been identified as luminescence from the stacking fault through the measurement of photoluminescence and cathodoluminescence via extensive spectroscopic and imaging techniques[13,14,42,50–54]. In contrast to the intrinsic S-series, the $D_4$-line exhibits distinctive characteristics, including the spatially localized emission profile at stacking faults, the relatively extended lifetime, and the excitation resonances below the bandgap. However, the exact spectral profile of the D-series considerably varies depending on the quality of the crystals, and the physical origins of individual features are attributed to not only different types of stacking faults[13,14,51,52] but also to boron-nitride divacancies[50].

**DUV laser excitation spectroscopy at 4 K.** To further elucidate the physical origins of the radiative recombination and the photocarrier excitation processes, we characterize our hBN crystals by employing DUV laser excitation spectroscopy of photoluminescence and photocurrent. The magnitude of the luminescence lines and photocurrent exhibit characteristic dependence on the laser excitation energy due to the unique absorption and carrier relaxation processes[55]. Compared to the conventional absorption measurement based on transmittance or reflectance, photoluminescence excitation (PLE) spectra are particularly informative for identifying the structural origin in the heterogeneous structures from the correlation between the

specific luminescence line and excitation resonances, as demonstrated for quantum wells[56] and carbon nanotubes[57]. Our efficient elimination scheme of the excitation laser residue (Fig. 1c) allows the laser micro-spectroscopy system to measure the PLE spectra for all the emission lines for the S- and D-series.

Figure 4c shows the 2D color map of the PLE spectra obtained from the hBN crystal of the device shown in Fig. 2a by scanning the laser excitation energy ($E_X$) across the bandgap with a constant laser power. The vertical axis, horizontal axis and color scale correspond to the photoexcitation energy, energy and intensity of luminescence signal, respectively. The purple dashed line in Fig. 4c covers the artifacts from the elastic scattering residue of the excitation laser. We observe the contrasting dependence of the S- and D-series on the laser excitation energy, while the luminescence lines in the same series exhibit similar excitation spectra. The S-series is completely absent in the photoluminescence spectra for $E_X < \sim 6$ eV while the D-series remains without any significant modification of the spectral shape, which is shown by the horizontal linecuts for $E_X = 6.07$ eV and $E_X = 5.97$ eV (blue and red solid lines in Fig. 4d, respectively). This dependence on the excitation energy is provided in detail by the vertical linecuts representatively for the $S_3$- and $D_4$-line (blue and red circles in Fig. 4e, respectively). The energy level of the indirect exciton (labeled as iX) in the pristine structure is marked with black dashed lines in Fig. 4d, e as a reference. The luminescence intensity of the S-series increases with the photoexcitation energy above 6 eV and exhibits the excitation resonance at 6.07 eV, which can be attributed to the combination of the direct and indirect phonon-assisted optical transitions in the pristine hBN structure[8,12,15,58]. In contrast, the luminescence intensity of D-series exhibits another excitation resonance at 5.97 eV below the excitation resonances observed for the S-series. Following the structural identification of the $D_4$-line by the cathodoluminescence imaging[13,14,52], we attribute the excitation resonance at 5.97 eV to the optical transitions in the stacking faults of our hBN crystals. The nearly identical PLE spectra for all the luminescence peaks of D-series indicate that the physical origin of $D_1$- and $D_2$-line can be also correlated to the same stacking fault inherent in our as-exfoliated crystals (Supplementary Fig. 8 for the vertical linecuts at the energies of other S- and D-lines).

The excitation resonances in the pristine structure and stacking faults of hBN also contribute to the photocarrier generation in the photocurrent spectrum (Fig. 4f) which is measured using the same heterostructure device in Fig. 2a with a bias voltage of 3 V. For excitation at 4.48 eV, far below the bandgap, the device exhibits negligible photocurrent of ~0.27 nA, which is presumably due to the optical transition from the deep trap states of hBN[25,27] or the photo-assisted tunneling across the vertical interface of the graphene electrodes and hBN[22]. For the excitation across the bandgap, the spectrum shows almost two orders of magnitude higher photocurrent with the excitation resonances at 5.97 and 6.07 eV, which are identically observed in our PLE spectra.

## Discussion

Although the exact absorption spectra cannot be determined due to the thicknesses of the hBN crystals herein (which is similar to the penetration depth)[44] and the lack of information on the carrier dynamics with different excitation energy[55], the emergence of the additional excitation resonance in our PLE and photocurrent spectra is a direct evidence of significant modification of the electronic structure in the stacking fault. This is consistent with the results of previous theoretical and experimental studies. The pristine structure of hBN single crystal exhibits the so-called AA' stacking order where boron (or nitrogen) in one layer is positioned exactly on top and bottom of nitrogen (or boron) in the adjacent layer[9]. First principle

calculations show that the deformation of the stacking order significantly modifies the electronic structure in hBN crystals, leading to the optical transition below the absorption edge of AA'-stacked hBN[14,59,60]. Reflection spectroscopy demonstrates the emergence of additional spectral features below 6 eV after the mechanical deformation[10].

Interestingly, the electroluminescence spectra exhibit additional spectral features that are not present in the photoluminescence spectra of the as-exfoliated hBN crystals. For example, Device A in Fig. 4a shows luminescence lines at 5.94 and 5.99 eV as well as a shoulder-like feature at the low energy side of the $S_4$-line. We note that the spatial profiles of electroluminescence (insets of Fig. 3d, e) are extended outside the overlapping area of graphene electrodes where the interface of the emissive hBN layer and the encapsulation hBN layers can form stacking structures other than those in the as-exfoliated hBN crystals. Indeed, recent cathodoluminescence studies demonstrate that the luminescence spectra can be dramatically modified by stacking angles in the vdW stack of two hBN crystals[10,11,13,14,51,52,54,61,62]. In addition, the strong electric field required for the carrier injection can modify the electronic structure of hBN. In the future, characterizing the local atomic structure of the stacking defects with excitation spectroscopy and understanding the effect of a strong electric field on their optical properties will provide valuable information to engineer DUV absorption and emission properties with stacking orders in hBN[14,59,60].

In conclusion, we reported DUV electroluminescence and photocurrent generation in vdW heterostructures where graphene electrodes inject or collect charge carriers in a photoactive hBN layer at DUV frequencies. Our laser excitation spectroscopy identified that the radiative recombination and the photocarrier generation processes in the heterostructure devices occur dominantly from the pristine structure and stacking faults of hBN. Although photocurrent generation is remarkably efficient, our hBN vdW heterostructures at room temperature exhibited low electroluminescence EQE (~$10^{-5}$ %) even for the energy range above 5.7 eV where achieving high EQE is fundamentally challenging for light-emitting diodes based on $Al_{1-x}Ga_xN$ (~$10^{-4}$%)[19,63–65]. However, we expect that electroluminescence EQE can be significantly improved by achieving a balanced carrier injection at low bias voltages, suppressing carrier leakage with blocking layers, and employing multiple emissive layers in the heterostructures, as demonstrated in the highly efficient light-emitting devices based on other inorganic semiconductors[19]. Our study reveals the potentials of hBN for efficient light-emitting and detecting devices at DUV and higher frequencies.

## Methods

**Fabrication of the vdW heterostructures of hBN**. The vdW heterostructures are prepared by the dry transfer technique[66]. Monolayer ~ trilayer graphene and 4~70 nm thick hBN crystals are separately exfoliated onto a silicon substrate with a 90 nm thick oxide layer. The thickness of graphene and hBN is first identified by the atomic force microscope (AFM). For the fabrication of the photocurrent devices, the thickness of graphene electrodes is varied from monolayer to trilayers. We use the polyethylene terephthalate (PET) stamp to pick up the Gr/hBN/Gr structure where the vdW materials are accurately aligned by a transfer stage based on an optical microscope. Then, the PET stamp with the heterostructure is stamped onto a 90 nm thick SiO₂/Si substrate with pre-patterned Ti/Au electrodes. The polymer and samples are heated at 70 °C for the pick-up process and 130 °C for the stamp process. The PET stamp is dissolved in dichloromethane for 12 h at room temperature. For fabricating the electroluminescence devices, monolayer graphene is used for the electrodes. The heterostructure is fabricated using the thermoplastic methacrylate copolymer (Elvacite 2552C, Lucite International) stamp to pick up the hBN/Gr/hBN/Gr/hBN structure for the electroluminescence devices with the same transfer stage. The Elvacite stamp is heated at 77 °C for the pick-up process and 170 °C for the stamp process. Thereafter, the stamp is dissolved in acetone for 1 h at room temperature. We use CF₄ plasma to partially etch the top hBN in order to expose graphene electrodes for the electrical contacts. Cr/Au electrodes are deposited by e-beam evaporation. The devices fabricated with the Elvacite stamp exhibit significantly less bubbles at the vdW interfaces than the devices fabricated with the PET stamp[67], which, we find, is helpful for improving the electroluminescence process.

**DUV luminescence and photocurrent spectroscopy**. THG and FHG of femtosecond laser pulses with 80 MHz repetition rate are utilized as widely tunable photoexcitation source. The laser is guided into a home-built confocal microscope constructed with DUV-enhanced Al mirrors, DUV-enhanced reflective objective, $CaF_2$ lens, and $CaF_2$ beam splitters. The laser beam spot on the sample is ~2 μm. The sample is held in a closed-cycle cryostat (Montana Cryostation s50) for DUV measurement at 10 K and room temperature. The sample chamber is at high vacuum to minimize any possible DUV-induced degradation. For resonant PLE spectroscopy, the laser residue is successfully blocked by a spatial filter, as described in the main text. For the sensitive detection of the luminescence spectra, we use a spectrometer equipped with an electrically-cooled Si CCD at Materials Imaging & Analysis Center of POSTECH. For electroluminescence and photocurrent spectroscopy, bias voltage is applied with a sourcemeter (Keithley 2400). The photocurrent is measured by the combination of the low noise current preamplifier (SRS SR570), the optical chopper, and the lock-in amplifier. The dark current is measured by the semiconductor parameter analyzer (Keithley 4200). Photocurrent mapping is enabled by the piezo-stages in the cryostat. I-V characteristics of the electroluminescence devices are measured with Keithley 2400.

## Data availability

The data that support the findings of this study are available from the corresponding author on request.

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

## Acknowledgements

The authors acknowledge helpful discussions with Jong Kyu Kim. J.K., S.H.Y., S.Y., S.-B.S. acknowledge the support from the National Research Foundation of Korea grants (NRF-2020R1A4A1018935, and 2020R1A2C2103166) and Samsung Electronics Co., Ltd.. This study was supported by Brain Korea 21 FOUR project for Education and research center for future materials (F21YY7105002). H.-W.J. and G.-H.L. acknowledge the support of National Research Foundation of Korea (NRF) funded by the Korean Government (Grant No. 2020R1C1C1013241). S.-Y. S., S.C. and M.-H.J. acknowledges the support from the Institute for Basic Science (IBS), under Project Code IBS-R014-A1. K.W. and T.T. acknowledge support from the Elemental Strategy Initiative conducted by the MEXT, Japan, Grant Number JPMXP0112101001, JSPS KAKENHI Grant Number JP20H00354 and the CREST(JPMJCR15F3), JST.

## Author contributions

J.K. conceived the project. S.-B.S., S.H.Y., S.Y.K., H.-W.J., G.-H.L., J.S.K., and S.Y. fabricated the vdW heterostructure devices of hBN crystals and obtained the DUV PLE spectra. S.-B.S., S.H.Y., S.-Y.S., S.C., and S.Y. measured the photocurrent and electroluminescence from the devices. J.K., M.-H.J., S.-B.S., S.H.Y., S.Y., and S.C. analyzed the experimental data. K.W. and T.T. grew the hBN crystals. All authors discussed and wrote the manuscript together.

## Competing interests

The authors declare no competing interests.
