## [Peer Review File · Nature Communications]

REVIEWER COMMENTS

Reviewer #1 (Remarks to the Author):

This paper represents the first report of deep-UV electroluminescence from graphene/hBN/graphene vdW heterostructures. DUV light emitting devices have received a lot of attention and this work certainly could provide a new direction. In the reviewer's opinion, however, the characterization of EL with current injection is not yet sufficient. Additional characterizations will enable readers/authors to understand the injection and recombination mechanisms. Specific comments are given below:

1. The authors should show both L-I and reverse I-V characteristics. Given the symmetry in band structure under zero bias (Fig 1b), did the authors observe light emission under reverse bias?
2. The authors assert in abstract of "strong electroluminescence", but the EQE is estimated to be less than 0.001%,
3. The authors may want to show EL spectrum as a function of injection current from 10 to 70 μ A,
4. In Fig 3d, why is there luminescence outside the graphene intersection area?
5. The agreement between EL and PL seems fortuitous as EL is generated under a very high bias with an intense local electrical field. It is a bit surprising that the electronic properties are not modified.
6. The authors showed in SI of three devices with different encapsulation with hBN, showing excellent improvement when the device is encapsulated from below and above. Can the authors comment on the reproducibility of their conclusion based on many samples?
7. An important study would be the dependence of EL on the thickness of the emissive hBN layer. Can these authors comment on the range of thickness that would support DUV EL?

Reviewer #2 (Remarks to the Author):

Song et al. reported hBN based electronic devices for light-emitting and photodetection application in the UV range. They demonstrated electroluminescence and spatial photocurrent mapping vertically stacking hBN/graphene heterostructures. I find it is hard to recommend this work for publication for the following:

1. The electroluminescence devices from hBN have been published in Nature Photonics (<https://www.nature.com/articles/nphoton.2009.167>) by Kenji Watanabe, Takashi Taniguchi et al. and recently by Chae et al. (https://doi.org/10.1364/CLEO_SI.2019.SM10.2). Thus, the argument about DUV EL from hBN in the abstract is wrong, which also suggests the authors are not familiar with the state-of-art research within this field.
2. This paper is not complete. There is not a conclusion section.
3. The detailed discussion is missing. It seems they focus on describing their results, which makes this paper isolate from the current research. I would suggest the authors place their results with others in the state to extract the significance of their work. For example, discuss the advantages of their devices in terms of energy efficiency and compactness.
4. Why they choose two devices to demonstrate EL and photodetection? I assume they use thicker hBN for photodetection devices because it has higher absorption, but it is not discussed in the paper. The other possibility is the devices have a short shelf time, as hBN would break down at a high electric field. Again this is not discussed anywhere in the paper.

5. What effects would the top hBN (encapsulation layer) have on the photocurrent?

Reviewer #3 (Remarks to the Author):

This manuscript reports deep-UV photocurrent and electroluminescence spectroscopy and imaging of hBN devices with graphene electrodes. The authors provide a comprehensive set of challenging measurements in the deep-UV, with much higher responsivity and external quantum efficiency of photocurrent and electroluminescence than earlier work in the near-UV. I find the claims in the manuscript to be well-supported by the data presented, the methodology is sound, and details are adequately provided. In my opinion this work is a significant advance over prior work on hBN photocurrent and electroluminescence generation and is therefore suitable for publication in Nature Communications.

Minor comments:

--the paper appears to lack a conclusion that summarizes the main results of the paper

--Fig. 3e might be more informative with a vertical axis that covers a smaller range

--Why were the EL measurements only conducted at 10 K? Does EL not occur at elevated temperatures? This point is important for potential applications of hBN for DUV generation.

--Fig. S1 convincingly makes the point that photocurrent is smaller for below-gap excitation, but this is expected anyway and is convincingly demonstrated in Fig. 4d already. Replotting S1 with a color scale that shows variations in the photocurrent as a function of position would be more informative to see whether differences appear with respect to above-gap excitation.

1. Reply to Reviewer #1

Original comment (1):

This paper represents the first report of deep-UV electroluminescence from graphene/hBN/graphene vdW heterostructures. DUV light emitting devices have received a lot of attention and this work certainly could provide a new direction. In the reviewer's opinion, however, the characterization of EL with current injection is not yet sufficient. Additional characterizations will enable readers/authors to understand the injection and recombination mechanisms. Specific comments are given below:

Our reply:

We deeply appreciate the referee for her/his effort to read our manuscript in this difficult period under the global pandemic of COVID-19. In addition, we appreciate for her/his appreciation of the importance of our work and valuable feedbacks on our manuscript. We have addressed her/his questions carefully as below.

Original comment (2):

1. The authors should show both L-I and reverse I-V characteristics. Given the symmetry in band structure under zero bias (Fig 1b), did the authors observe light emission under reverse bias?

Our reply:

We appreciate the reviewer for her/his valuable suggestions to further characterize our electroluminescence devices. Following the reviewer's suggestion, we show L-I and I-V characteristics both for the forward and the reverse bias voltages in Fig.3f and 3b, respectively in the revised manuscript. In addition, we have provided the detailed analysis on their characteristics in the revised main text highlighted with red colors.

We do observe light emission under the reverse bias. Despite the symmetry of energy band diagram under zero bias, I-V characteristics, electroluminescence EQE, and electroluminescence spatial profile in Fig.3 in the revised manuscript show notable discrepancy under the two bias polarities. We show that the discrepancy originates from the asymmetric charge carrier injection and transport depending on the exact shape of the electrodes and the bias polarities. First, I-V characteristics indicates that the tunneling current is dominated by a single type of charge carriers under both bias polarities (Fig.3c in the revised manuscript). We attribute holes as major charge carriers based on the recent study on the Fowler-Nordheim tunneling injection at the interface of graphene and hBN [ACS Applied Materials & Interfaces 10, 11732-11738 (2018), Nature Communications 12, 1000 (2021)]. Electron injection occurs but provides significantly less contribution to the total current due to higher tunneling barrier height than holes. The detailed analysis (Fig.3c in the revised manuscript) based on Fowler-Nordheim tunneling mechanism reveals that tunneling current of holes is already asymmetric due to formation of charged traps [PRB 97, 045425 (2018)] or different effective tunneling areas of anodes under the forward and the reverse bias voltages. Second, the spatial profile of electroluminescence captures the localized areas where carrier injection is relatively more efficient. Insets of Fig.3d and 3e in the revised manuscript show that the electroluminescence signal appears particularly strong at the corners or the edges of the area overlapping two graphene electrodes. This is due to enhancement of local electric field at the boundaries of electrodes [Light-Emitting Diodes. 2 edn (Cambridge University Press, 2006)] which facilitates tunneling carrier injection. This behavior is generically observed from most of our devices (Fig.S6 in the revised supplementary information) Therefore, the specific shape of electrodes, the bias polarities and the quality of hBN crystal can modify tunneling carrier injection and radiative recombination which determine electrical and light emission characteristics. We have included the discussion above in the revised manuscript.

Original comment (3):

2.The authors assert in abstract of "strong electroluminescence", but the EQE is estimated to be less than 0.001%.

Our reply:

We appreciate the reviewer for her/his critical comment to compare our work in the broader context for light emitting device applications. Following the reviewer's suggestion, we have removed the word "strong" in the abstract. We agree that the electroluminescence EQE of our devices must be improved by orders of magnitude in order to be used for any practical usage yet.

Original comment (4):

3. The authors may want to show EL spectrum as a function of injection current from 10 to 70 μA .

Our reply:

We appreciate the reviewer for her/his helpful suggestion to significantly improve the presentation of our figures in the manuscript. Following the reviewer's suggestion, we have shown electroluminescence spectra gradually from 10 μA to 40 μA in Fig.3d and 3e of the revised manuscript. As the current increases, intensity of electroluminescence lines monotonically increases while their spectral profile maintains nearly identical.

Original comment (5):

4. In Fig 3d, why is there luminescence outside the graphene intersection area?

Our reply:

We appreciate the reviewer for her/his insightful question on the spatial profile of electroluminescence in our devices. In order to address the reviewer's question, we have fabricated more devices and carefully examined the spatial profiles of electroluminescence (Insets of Fig.3d and 3e and Fig.S6). The electroluminescence signal is mostly concentrated in or near the area overlapping two graphene electrodes, but the signal appears particularly strong at the corners or the edges of the area. This is due to enhancement of local electric field at the boundaries of electrodes [Light-Emitting Diodes. 2 edn, (Cambridge University

Press, 2006)], which facilitates carrier injection via Fowler-Nordheim tunneling. This strong electric field at the boundaries also spreads charge carriers horizontally outside the overlapped area which gives electroluminescence signals [Light-Emitting Diodes. 2 edn, (Cambridge University Press, 2006)]. We have included the discussion above in the revised manuscript.

Original comment (6):

5. The agreement between EL and PL seems fortuitous as EL is generated under a very high bias with an intense local electrical field. It is a bit surprising that the electronic properties are not modified.

Our reply:

We appreciate the reviewer for her/his important comments which has encouraged us to further characterize electroluminescence spectra in comparison to photoluminescence spectra. Following the reviewer's comment, we have fabricated 10 more devices and carefully examined their electroluminescence spectra at a temperature of 10 K (top panel of Fig.4a of the revised main text and Fig.S2 of the revised supplementary information). Significant disagreement has been indeed observed between electroluminescence and photoluminescence spectra. Although the emission lines in photoluminescence spectra (S-series and D-series) in bottom panel of Fig.4a of the revised main text are also observed in the electroluminescence spectra, relative ratios of their intensity differ among different electroluminescence devices. Furthermore, the electroluminescence spectra show additional features which are absent in photoluminescence spectra from our as-exfoliated hBN crystals. For example, red solid line in Fig.4a shows luminescence lines at 5.94 eV and 5.99 eV and a shoulder-like feature at the lower energy side of S₄-line.

We suspect two possible origins for disagreement between electroluminescence and photoluminescence spectra. First, as the reviewer questioned, the light emission properties and the electronic structure of hBN can be modified by strong electric field. Carrier injection requires electric field typically over 0.5 V/nm which is substantially strong to modify the interlayer interaction of vdW materials [Nature 459, 820–823 (2009)]. Also, strong electric field can broaden luminescence lines due to exciton ionization [Physical Review B 13, 12 (1976)]. Second, charge carriers can emit luminescence lines in hBN with different stacking orders which do not exist in our as-exfoliated hBN crystals. The spatial profiles of electroluminescence (for example, insets of Fig.3d and 3e in the revised manuscript) are extended outside the area overlapping graphene electrodes due the strong electric field. The interface of the emissive hBN layer and the encapsulation hBN layers can form stacking structures other than the AA' stacking order as reported by recent studies

[ACS Photonics 1, 857-862 (2014)]. Indeed, the cathodoluminescence studies demonstrate that luminescence spectra can be dramatically modified by stacking orders [Physical Review B 89, 035414 (2014), ACS Photonics 1, 857-862 (2014), arXiv:2108.04747 (2021), Applied Physics Letters 89, 141902 (2006), J. Appl. Phys. 102, 116102 (2007), Diamond and Related Materials 20, 849-852 (2011), Nano Letters 21, 2832-2839 (2021), Nature Materials 3, 404-409 (2004), Int. J. Appl. Ceram. Technol., 8, 977–989 (2011)].

The exact origin can be investigated by more detailed photoluminescence measurement on the hBN devices with control of the electric field in-situ and with control of the stacking order. Unfortunately, this is beyond the scope of the current work, but this is certainly of our interest in the future. We have added a paragraph at the end of the revised main text to provide the discussion above.

Original comment (7):

6. The authors showed in SI of three devices with different encapsulation with hBN, showing excellent improvement when the device is encapsulated from below and above. Can the authors comment on the reproducibility of their conclusion based on many samples?

Our reply:

We appreciate the reviewer for her/his suggestion to share electroluminescence spectra on extended number of samples, which can strengthen our conclusion on the effect of hBN encapsulation. Following the reviewer's suggestion, we have improved the reproducibility of our conclusion by fabricating 12 more devices. We now present electroluminescence spectra from 14 devices with different structures. Fig.S1a and S1b in the revised supplementary information are spectra from 2 devices without any supporting or capping hBN layer at all, which shows broad electroluminescence spectra without any well-defined emission lines from S-series and D-series. Fig.S1c and S1d in the revised supplementary information are spectra from 2 devices only with a supporting layer at the bottom, which shows emission lines similar to D-series while emission lines from S-series is absent or

broadened dramatically. On the other hand, 10 devices with hBN encapsulation (Fig.S2 in the revised supplementary information) shows emission lines from both S-series and D-series. Therefore, although the exact spectra vary for the reasons we discussed above (original comment (6)), we believe it is true that encapsulation does significantly improve the electroluminescence process. We have included the additional data above in the revised supplementary information.

Original comment (8):

7. An important study would be the dependence of EL on the thickness of the emissive hBN layer. Can the authors comment on the range of thickness that would support DUV EL?

Our reply:

We appreciate the reviewer for her/his critical question on the range of hBN thicknesses which support DUV electroluminescence. Following the reviewer's suggestion, we have investigated devices with different thicknesses of the emissive hBN layer ranging from 4 nm to 70 nm. Fig.S2 presents electroluminescence spectra at a temperature of 10 K. The insets show the device images and the thickness of hBN emissive layers. The devices exhibit electroluminescence EQE randomly scattered from $\sim 10^{-5}$ % to $\sim 10^{-4}$ % without any clear thickness dependence. However, we would expect that few-nm thick hBN emissive layers results in significantly decreased EQE and low device stability. Contribution from direct tunneling process between two graphene electrodes exponentially increases as the hBN thickness gets thinner [*Applied Physics Letters* **99**, 243114 (2011)]. In addition, thinner hBN is prone to the electrical breakdown under high electric field. On the other hand, although thick hBN supports DUV electroluminescence, the device requires significantly higher voltage for 70 nm thick hBN, which will compromise the wall plug efficiency [Light-Emitting Diodes. 2 edn, (Cambridge University Press, 2006)]. We have included the discussion above in the revised supplementary information.

2. Reply to Reviewer #2

Original comment (1):

Song et al. reported hBN based electronic devices for light-emitting and photodetection application in the UV range. They demonstrated electroluminescence and spatial photocurrent mapping vertically stacking hBN/graphene heterostructures. I find it is hard to recommend this work for publication for the following:

Our reply:

We deeply appreciate the referee for her/his effort to read our manuscript in this difficult period under the global pandemic of COVID-19. In addition, we appreciate for her/his valuable feedbacks on our manuscript. We have addressed her/his questions carefully as below.

Original comment (2):

1. The electroluminescence devices from hBN have been published in Nature Photonics (<https://www.nature.com/articles/nphoton.2009.167>) by K. Watanabe, T. Taniguchi et al. and recently by Chae et al. (https://doi.org/10.1364/CLEO_SI.2019.SM1O.2). Thus, the argument about DUV EL from hBN in the abstract is wrong, which also suggest the authors are not familiar with the state-of-art research within this field.

Our reply:

We appreciate the reviewer for her/his critical question to justify the novelty of our report on DUV electroluminescence from vdW heterostructures of hBN. The advance in this work is the first demonstration and the detailed characterizations of DUV “electroluminescence” from the intrinsic bandgap and stacking faults from hBN. The first work [Nature Photonics 3, 591 (2009)] by K. Watanabe and T. Taniguchi, the reviewer has mentioned, reports the device demonstration which employs “cathodoluminescence” to emit DUV luminescence from hBN. This device requires a portable high voltage (> 1 kV) field emitter which excites electrons in the valence band to the conduction band. “Electroluminescence” is

fundamentally different electronic process in terms of charge carrier generation. In the electro-luminescence process, electrons and holes are created by carrier injection from the electrodes by applying moderate voltages as in our case.

On the other hand, CLEO_SI.2019.SM1O.2 by Chae *et. al.* reports the electroluminescence process from vdW heterostructures of hBN as in our case. However, the luminescence lies in the near ultraviolet (NUV) range (3 ~ 4 eV) from the deep-level trap states in hBN. In our work, we report the luminescence in the energy range from 5.4 eV ~ 6 eV. Furthermore, utilizing DUV laser excitation spectroscopy, we provide direct evidences that DUV light emission originates from the intrinsic bandgap and the stacking faults from hBN. Therefore, in our best knowledge, we believe that the previous studies have not reported DUV electroluminescence yet from the bandgap of hBN.

We have included the two works above in the revised reference list, and clarified the novelty of this work in the revised main text.

Original comment (3):

2. This paper is not complete. There is not a conclusion section.

Our reply:

We appreciate the reviewer for her/his effort to read our manuscript carefully. In the revised main text, we have now included a paragraph for the conclusion.

Original comment (4):

3. The detailed discussion is missing. It seems they focus on describing their results, which makes this paper isolate from the current research. I would suggest the authors place their results with others in the state to extract the significance of their work. For example, discuss the advantages of their devices in terms of energy efficiency and compactness.

Our reply:

We appreciate the reviewer for her/his important suggestion to emphasize the significance of our work by comparing to other existing works in the research field. As discussed above for original comment (2), we believe that DUV electroluminescence from hBN has not been reported yet previously. Therefore, unfortunately, we could not find any other DUV electroluminescence device based on hBN in order to deduce the advantages of our devices as the reviewer suggest. Instead, we have compared our work to DUV light emitting diodes (LED) based on other inorganic materials such as $\text{Al}_{1-x}\text{Ga}_x\text{N}$ in the revised main text. Electroluminescence EQE of our devices is comparable or slightly lower than the state-of-the-art $\text{Al}_{1-x}\text{Ga}_x\text{N}$ LED for the energy range above 5.7 eV where high EQE is fundamentally challenging to achieve for the light emitting diodes based on $\text{Al}_{1-x}\text{Ga}_x\text{N}$ [Appl. Phys. Lett. 96, 221110 (2010),. Appl. Phys. Express 12, 012008 (2019), Nature Photonics 13, 233-244 (2019), Light: Science & Applications 10, 129 (2021)]. However, hBN-based DUV LED holds promising potential because photon extraction from hBN emissive layers can be fundamentally more efficient than $\text{Al}_{1-x}\text{Ga}_x\text{N}$ due to the emission direction strongly oriented along the crystalline c-axis [Nature Reviews Materials 4, 552-567 (2019)]. We expect to improve electroluminescence EQE significantly higher by achieving balanced carrier injection at lower bias voltages, suppressed carrier leakage with blocking layers, and multiple emissive layers in the heterostructures as demonstrated in highly efficient light emitting devices based on other inorganic semiconductors [Nature Photonics 13, 233-244 (2019)]. We have included the discussion above in the revised main text.

Original comment (5):

4. Why they choose two devices to demonstrate EL and photodetection? I assume they use thicker hBN for photodetection devices because it has higher absorption, but it is not discussed in the paper. The other possibility is the devices have a short shelf time, as hBN would break down at a high electric field. Again this is not discussed anywhere in the paper.

Our reply:

We appreciate the reviewer for her/his important question to clarify the reason for choosing two different devices for the electroluminescence and photocurrent measurement. Two separate devices are used in our study because different device structures are required to properly study the photocurrent generation and the electroluminescence processes. For the photocurrent generation process, the capping hBN layer substantially absorbs incident DUV light, which prevents us to measure photocurrent in the active hBN layer. In order to address the reviewer's comment, we have fabricated the photocurrent device with a 10-nm thick hBN capping layer and measured the photocurrent as shown in Fig. R1 below. The thicknesses of active hBN layers are the same (10 nm). Without the capping layer, we observe strong photocurrent (61 nA) under the bias voltage of 1 V (blue solid line). However, the device shows only negligible photocurrent with the capping layer (red solid line).

For the electroluminescence process, on the other hand, the capping hBN layer, together with the supporting hBN layer, provides dramatically improved DUV luminescence spectra from the active hBN layer as we showed in Fig. S2 the revised supplementary information. The radiative recombination of electrons and holes requires significantly higher quality of vdW heterostructure with atomically clean and less-strained interfaces over macroscopic overlapping areas of two graphene electrodes. We have included the discussion above in the revised main text.

Fig. R1. I-V characteristics in the dark and under laser excitation with and without hBN encapsulation. The thickness of active hBN layers are the same (10 nm) for the device without the capping hBN and the device with the capping hBN. Without the capping layer, we observe strong photocurrent (61 nA) under the bias voltage of 1 V (blue solid line). However, the device shows only negligible photocurrent with the capping layer (red solid line). I-V characteristics of photocurrent are measured under an excitation power of 2.5 μ W at 6.22 eV.

Furthermore, in order to address the reviewer's concern on the stability of our electroluminescence devices, we have carried out extra measurement to elucidate the effect of high electric field on the hBN emissive layer from more systematic I-V characteristics and electroluminescence measurement. As shown in Fig.3 of the revised manuscript, the devices operate stable even at room temperature, which allows us to measure the detailed characteristics of the electroluminescence process including I-V characteristics, electroluminescence spectra, and electroluminescence imaging. However, indeed, we do observe that strong electric field forms defects and degrades electrical property of the emissive hBN layer. After multiple cycles under high bias voltages, the device shows significant amount of current under low bias voltages at room temperature (Fig.S5). However, our devices do not require extreme high voltages which can completely break the hBN emissive layer. We have discussed above detailed in the revised manuscript.

Fig. S5 I-V characterization via Poole-Frenkel emission mechanism at room temperature. **a.** I-V characteristics before defect formation (black solid line) and after defect formation (red solid line) at room temperature **b.** I-V characteristic of red solid line in (a) replotted with respect to $\ln(I/V_B)$ and $V_B^{1/2}$.

Original comment (6):

5. What effects would the top hBN (encapsulation layer) have on the photocurrent?

Our reply:

We appreciate the reviewer for her/his helpful question to clarify the effect of capping hBN on photocurrent. As discussed above for original comment (5), the capping hBN layer substantially absorbs incident DUV light. Even with a 10-nm thick hBN capping layer, photocurrent decreases by 300 times as shown in Fig.R1 above. We have provided the discussion above in the revised main text.

2. Reply to Reviewer #3

Original comment (1):

This manuscript reports deep-UV photocurrent and electroluminescence spectroscopy and imaging of hBN devices with graphene electrodes. The authors provide a comprehensive set of challenging measurements in the deep-UV, with much higher responsivity and external quantum efficiency of photocurrent and electroluminescence than earlier work in the near-UV. I find the claims in the manuscript to be well-supported by the data presented, the methodology is sound, and details are adequately provided. In my opinion this work is a significant advance over prior work on hBN photocurrent and electroluminescence generation and is therefore suitable for publication in Nature Communications.

Our reply:

We deeply appreciate the referee for her/his effort to read our manuscript in this difficult period under the global pandemic of COVID-19. In addition, we appreciate for her/his appreciation of the importance of our work and valuable feedbacks on our manuscript. We have addressed her/his questions carefully as below.

Original comment (2):

Minor comments:

--the paper appears to lack a conclusion that summarizes the main results of the paper

Our reply:

We appreciate the reviewer for her/his effort to read our manuscript carefully. In the revised main text, we have now included a paragraph for the conclusion.

Original comment (3):

--Fig. 3e might be more informative with a vertical axis that covers a smaller range

Our reply:

We appreciate the reviewer for her/his valuable suggestion to improve our figures. We have followed the referee's suggestion and changed the range of vertical axis much smaller (Fig.3f in the revised manuscript). In addition, we have shown the total integrated intensity of electroluminescence (L) versus current in the linear scale (the inset in Fig.3f of the revised manuscript) which can be more informative to the readers.

DUV electroluminescence spectra for current from 10 μA to 40 μA under the forward and the reverse biases, respectively. The spatial profiles of DUV electroluminescence are shown in the insets. **f.** Current dependence of electroluminescence EQE. The inset shows a plot of the total-integrated luminescence intensity (L) from 5.4 eV to 6 eV versus current. Red empty and filled circles correspond to the forward and the reverse biases, respectively.

Original comment (4):

--Why were the EL measurements only conducted at 10 K? Does EL not occur at elevated temperatures? This point is important for potential applications of hBN for DUV generation.

Our reply:

We deeply appreciate the reviewer for her/his critical question to improve significance of our work. Although our previous devices fabricated by PET stamps have operated stable at a temperature of 10 K, most of the devices have exhibited unstable behavior at room temperature. Under a given bias voltage, current often suddenly increases or decreases in the devices, which prevented to systematically investigate the electroluminescence process at an elevated temperature. The recent studies have reported similar behavior which is attributed to conduction channels via defects in hBN [Phys. Rev. B 97, 045425 (2018)].

In order to address the reviewer's question, we have tried to minimize possible defects created during the fabrication procedure of vdW heterostructures. New devices in the revised manuscript are fabricated with the thermoplastic methacrylate copolymer stamp which is helpful to fabricate vdW heterostructures with less-strained and atomically cleaner interfaces over macroscopic area overlapping two graphene electrodes. As a consequence, we could achieve the hBN devices which operates stable at room temperature. We have presented the detailed electrical and light emission characteristics in Fig.3 of the revised manuscript.

Original comment (5):

--Fig. S1 convincingly makes the point that photocurrent is smaller for below-gap excitation, but this is expected anyway and is convincingly demonstrated in Fig. 4d already. Replotting S1 with a color scale that shows variations in the photocurrent as a function of position would be more informative to see whether differences appear with respect to above-gap excitation.

Our reply:

We appreciate the reviewer for her/his insightful comment to check our experimental data to further understand the photocurrent generation process in hBN. Following the reviewer's suggestion, we have changed the color scale in order to show variations in the photocurrent as a function of position. Indeed, we do observe spatial inhomogeneity in the mapping image for below-gap excitation. (Black and white circles of Fig.S3c in the revised supplementary information). For the black circles, the mapping image for above-gap excitation also shows a similar spatial profile (Fig.S3b in the revised supplementary information). In the optical microscope image (Fig.S3a in the revised supplementary information), we find bubbles trapped at the interfaces of vdW heterostructures (circles with black dashed line) which can partially obstruct collection of charge carriers.

On the other hand, the area marked with white dashed circle shows relatively weaker photocurrent only for the below-gap excitation. However, we do not find any notable feature in the optical image at the same location. This particular inhomogeneity is potentially due to inhomogeneous density of defects which contribute to generate photocurrent for the laser excitation at 4.48 eV. It will be interesting to measure photoluminescence spectra as a function of position to correlate the actual defect density and the photocurrent magnitude. We have provided the revised figure and the discussion above in the revised manuscript.

REVIEWER COMMENTS

Reviewer #1 (Remarks to the Author):

This reviewer that the authors have satisfactorily addressed the his/her concerns and comments. The additional data and analysis provided by the authors greatly improved the quality of the paper. Among the many published papers regarding deep-UV LEDs, this paper fell much below in performance and device maturity. However, the use of 2D layers to induce DUV EL is novel enough in 2D device physics to warrant a publication.

Reviewer #2 (Remarks to the Author):

The author has provided more experimental results in this round, however, the results raised new questions from my perspective. Thus I am still confused and cannot recommend this paper to be published. Some of the points are listed below, I hope the editors' office and authors would take them into consideration.

1. The spatial distribution of EL. Under reverse bias, there is a clearly new EL region emerging in the bottom/top graphene overlap area, shown in fig 3e, fig s6g, s6h and s6i. The author completely ignored and no discussion was found.

And compared with the peak maximum in each plot (fig 3d and e), why does the peak at 5.5eV decrease at a reverse bias?

The author only discussed the F-N tunnelling which enables the emission at the edge, which I also wouldn't connect the dots. Should the electric field be the same across the overlapping region? Why does new EL emission appear in some regions but not the other under reverse bias?

2. The effect of hBN capping.

The author claimed the capping would help to improve the emission quality, this is not substantiated. In fig s2, the spectra are all random and did not change over time. So there is no sign of emission gets better.

Consider the voltage applied, the electric field should be high enough to break down hBN materials. Thus, I assume part of these EL lines would be exciton from other materials. Thus, correlated PL and EL measurements are important to determine the EL emission lines are excitonic emission from hBN.

Fig r1, why would a 10 nm hBN capping layer would reduce the photocurrent to none? This is not discussed, I assume this is due to significant light absorption in the DUV range by the 10nm capping layer. But thicker capping layer on EL devices does not seem to reduce the emission, which I find odd.

3 EL intensity. I would expect to see the relationships between intensity and current, intensity and layer thickness. This is due to the charge injection efficiency and thickness of the emission layers. The intensity shall increase with the injection and injection decreases with the thickness. But in fig s2, they are all random, why is that case?.

4. Some minor points.

Experimental details and extracted results shall be shown where is handy (e.g. figure captions) for readers to put information together. I find really hard to correlate the parameters with results. For

example, EL mapping in fig s6, emission in what spectral range is collected? What are the thicknesses for the hBN capping layers in fig s2? what laser is used to excite the device in fig r1, and so on?

Reviewer #3 (Remarks to the Author):

In my opinion the revised manuscript has been significantly improved, and the authors have responded satisfactorily to all of my comments. I therefore recommend publication of this work.

1. Reply to Reviewer #1

Original comment (1):

This reviewer that the authors have satisfactorily addressed the his/her concerns and comments. The additional data and analysis provided by the authors greatly improved the quality of the paper. Among the many published papers regarding deep-UV LEDs, this paper fell much below in performance and device maturity. However, the use of 2D layers to induce DUV EL is novel enough in 2D device physics to warrant a publication.

Our reply:

We sincerely appreciate the referee for her/his appreciation of the improvements in our revised manuscript and for her/his recommendation on our work for the publication in Nature Communications.

2. Reply to Reviewer #2

Original comment (1):

The author has provided more experimental results in this round, however, the results raised new questions from my perspective. Thus I am still confused and cannot recommend this paper to be published. Some of the points are listed below, I hope the editors' office and authors would take them into consideration.

Our reply:

We sincerely appreciate the referee for her/his effort to read our revised manuscript and provide valuable comments on the experimental results. We have addressed her/his questions carefully as below.

Original comment (2):

1. The spatial distribution of EL. Under reverse bias, there is a clearly new EL region emerging in the bottom/top graphene overlap area, shown in fig 3e, fig s6g, s6h and s6i. The author completely ignored and no discussion was found.

Our reply:

We appreciate the reviewer for her/his question on the spatial profiles of electroluminescence, which is helpful to improve the presentation of our manuscript. The discussion on the different spatial profiles was provided with the additional data and analysis (Fig.S6) in the supplementary information 5 of our previous manuscript. In the revised manuscript, we have added a sentence in the main text which helps guide readers to find the discussion in the supplementary information. Furthermore, we have provided the additional spatial profiles of electroluminescence at 10 K (Fig.S7) and related discussion below in the revised supplementary information, which describes the observed behavior more clearly.

We find that electroluminescence signal (insets of Fig.3d and 3e, and also Fig.S6 and S7) appears particularly strong at the corners or the edges of the area overlapping two graphene electrodes. As the reviewer points out, the devices obviously exhibit the different spatial profiles under the forward and the reverse bias. This behavior can be understood from tunneling carrier injection of electrons and holes at the junction of hBN and graphene. First of all, the electric field is “not” uniform across the overlapping area of graphene electrodes. The electric field is typically enhanced at the fringes of electrodes, which is well-known phenomena and is also often discussed detailed, for example, in the textbook for light-emitting devices [Schubert, E. F. Light-Emitting Diodes. 2 edn, p.133-136 (Cambridge University Press, 2006)]. This electric field enhancement plays relatively a less important role for LEDs with ohmic contacts. For our devices, however, the carrier injection rate is extremely sensitive to this enhanced electric field since carriers are injected via Fowler-Nordheim tunneling mechanism. Second, the hole injection is more efficient than the electron injection at the interface of graphene and hBN due to the smaller barrier height as reported by the recent studies [ACS Applied Materials & Interfaces 10, 11732-11738 (2018), Nature Communications 12, 1000 (2021)]. As the bias voltage increase, initially only holes are

injected along the edges of anodes where the electric field is enhanced. Electrons are not injected since higher electric field is required to inject electrons. Under higher bias voltages, holes are injected and spread through relatively wider area in the hBN emissive layer while electrons start getting injected mostly through the edges of cathodes. Therefore, holes and electrons recombine and luminesce preferably appear along the edges of cathodes in the overlap area. This behavior can be observed clearly observed in the spatial profile at 10 K (Fig.S7). The red dashed lines and the blue dashed lines correspond to the edges of the cathodes and the anodes in the devices, respectively. As flipping the bias polarity, the electroluminescence signal emerges at the edges of cathodes in the overlap areas. In comparison to the spatial profile at room temperature (Fig.S6), overall signal is strong at 10 K, which helps to visualize the spatial profile more clearly. Nevertheless, the spatial profiles at room temperature also qualitatively show similar behavior.

However, we agree that this simple explanation cannot capture all the features of the spatial profiles. For example, the inset of Fig.3e shows the signal at the center of the overlap area under the reverse bias, which is absent under the forward bias (the inset of Fig.3d). In addition, not all the edges of cathodes show electroluminescence signals. In order to explain this, we have to consider defects at the interfaces of van der Waals structures as well as defects within hBN crystals. As discussed in the literature [Schubert, E. F. Light-Emitting Diodes. 2 edn, (Cambridge University Press, 2006)], these defects can alter the flow of carriers and also trap carriers at the localized areas, which acts differently depending on the carrier polarity and temperature. In the future, more detailed spatial characterization of defects in the devices will help to address this issue, which unfortunately is beyond the scope of this study.

Original comment (3):

And compared with the peak maximum in each plot (fig 3d and e), why does the peak at 5.5 eV decrease at a reverse bias?

Our reply:

We appreciate the reviewer for her/his effort to carefully check our electroluminescence spectra under the two bias polarities. As discussed in detail in our main text with Fig.4, the previous works [Physical Review B 89, 035414 (2014), ACS Photonics 1, 857-862 (2014), Journal of Applied Physics 103, 103520 (2008), Physical Review B 93, 035207 (2016), Applied Physics Letters 89, 141902 (2006), J. Appl. Phys. 102, 116102 (2007), AIP Advances 10, 075025 (2020), Diamond and Related Materials 20, 849-852 (2011)] and also our photoluminescence excitation spectroscopy identify the origin of the peak at 5.46 eV (the peak referred as the peak 5.5 eV by the reviewer) as luminescence from the stacking faults which can be distributed inhomogeneously in the emissive hBN layer. The red solid line in the top panel of Fig.4a is electroluminescence spectrum at 10 K from the same device in Fig.3. The peak at 5.46 eV corresponds to D₄-line in the photoluminescence spectrum (the bottom panel of Fig.4a), which is thoroughly studied in the previous literature and is identified as luminescence from the stacking faults in hBN crystals. Therefore, the reverse bias can emit smaller signal for D₄-line than the forward bias case because the electroluminescence occurs in the area with relatively smaller density of stacking faults. The fact that electroluminescence show different spatial profiles under two bias polarities is also consistent with this explanation.

Original comment (4):

The author only discussed the F-N tunnelling which enables the emission at the edge, which I also wouldn't connect the dots. Should the electric field be the same across the overlapping region? Why does new EL emission appear in some regions but not the other under reverse bias?

Our reply:

We appreciate the reviewer for her/his question on the reason for the spatial profiles of electroluminescence emerging along the edges of the overlapping region. We have addressed this issue by providing detailed discussion above for **Original Comment (2)**.

Original comment (5):

2. The effect of hBN capping.

The author claimed the capping would help to improve the emission quality, this is not substantiated. In fig s2, the spectra are all random and did not change over time. So there is no sign of emission gets better.

Our reply:

We appreciate the reviewer for her/his comment to clarify the effect of hBN encapsulation on the emission spectra. As discussed in our previous manuscript and supplementary information 1, the improvement of emission quality was compared for the two cases: with encapsulation and without encapsulation. Following the comments from the reviewer 1 in the previous round, we have fabricated 10 more devices and measured electroluminescence spectra (Fig.S1 and S2). Overall, the encapsulated devices (Fig.S2) show prominent luminescence lines both from S-series and D-series while the emission lines are absent or significantly broadened in the devices without the encapsulation (Fig.S1). In this sense, we claim the hBN capping indeed improves the emission quality.

However, as the reviewer points out, it is true that the detailed features in electroluminescence spectra vary from devices to devices even with the hBN encapsulation. This means that hBN encapsulation cannot solve all the problems completely. On the other hand, we would like to note that hBN encapsulation cannot resolve the “internal” defect issues even for graphene or few-layers of transition metal dichalcogenides (TMD) although hBN encapsulation successfully resolves issues from the “external” environment [Nature Materials 18, 541-549 (2019)]. We observe similar behavior for our hBN devices as well. For example, the devices in Fig.S2 show different intensity ratio for S-series and D-series. This can imply that the emissive hBN layers (~ few tens of nanometers) possess different density of stacking faults which was created during the crystal growth or the device fabrication processes. We believe that this issue also should be successfully resolved if one uses higher quality of hBN bulk crystals to start

from or fabricate vdW heterostructures more carefully to avoid the formation of interfacial bubbles or strain, which will be of our future interest. We have added the discussion above in the revised Supplementary Information 1.

various active hBN layer thicknesses at 10K. **a.** 4nm. **b.** 7nm. **c.** 8nm. **d.** 10nm. **e.** 14nm. **f.** 15nm. **g.** 34nm. **h.** 36nm. **i.** 50nm. **j.** 70nm Insets show optical microscope images of the devices for electroluminescence measurement. Black dashed lines indicate graphene electrodes. The scale bar corresponds to 15 μm . The hBN capping layers have the thickness range from 10 nm to 50 nm.

Original comment (6):

Consider the voltage applied, the electric field should be high enough to break down hBN materials. Thus, I assume part of these EL lines would be exciton from other materials. Thus, correlated PL and EL measurements are important to determine the EL emission lines are excitonic emission from hBN.

Our reply:

We understand the reviewer's concern on the fairly high electric field required for our hBN devices. However, as we discussed in the previous reply [Our response to **Original comment (5)** in the previous reply], our detailed electrical and optical characterization (Fig.3 and 4) show that the devices operate properly emitting electroluminescence lines from the pristine structure and the stacking faults of hBN. Our apologies if our explanation was not clear in the previous reply. First of all, we would like to remind that our devices consist of only graphene and hBN layers (Fig.1a and 3a). We have only hBN layers between graphene electrodes, and we do not have any other materials. Graphene cannot luminesce at DUV frequency since it is a semiconductor with zero bandgap.

Furthermore, just like the reviewer's suggestion, our previous manuscript had indeed provided detailed optical characterization at 10 K to correlate PL and EL spectra (Fig.4a). Red solid line in the upper panel (labelled as Device A) shows electroluminescence spectrum at a temperature of 10 K from the same device in Fig.3. In comparison to the spectra at room temperature, additional sharp spectral features are revealed as the width of luminescence lines becomes narrower overall at lower

temperature. We find that other devices (labelled as Device B and C) share qualitatively similar features in electroluminescence spectra. Black solid line in the bottom panel shows the representative photoluminescence spectrum commonly observed from the as-exfoliated hBN crystals with the laser excitation energy of 6.22 eV. Photoexcited carriers via the band-to-band transition exhibit series of emission lines which are also present in electroluminescence spectra with good agreement. According to the previous studies [Nature Photonics 10, 262-266 (2016), Nature Reviews Materials 4, 552-567 (2019), Physical Review Letters 122, 067401 (2019), Physical Review Letters 122, 187401 (2019), Physical Review B 99, 081109 (2019)], the narrow four lines, referred as S-series, at 5.89 eV (S₁-line), 5.86 eV (S₂-line), 5.79 eV (S₃-line), and 5.76 eV (S₄-line) originate from intrinsic indirect exciton at ~ 5.96 eV (labelled as iX marked with black dashed line) in the pristine structure of hBN crystals via emission of TA, LA, TO, and LO phonons at T point, respectively, in the momentum space, which compensates the momentum mismatch for the conversion of indirect exciton to photon in the free space. On the other hand, the relatively broad lines below 5.7 eV, referred as D-series, including the peaks at 5.63 eV (D₁-line), 5.56 eV (D₂-line), 5.47 eV (D₄-line) and the shoulder feature at 5.5 eV (D₃-line) are dominantly inherent from stacking defects in hBN crystals, which are analyzed with our photoluminescence excitation spectroscopy (Fig.4b-4d) and the previous literature [Physical Review B 89, 035414 (2014), ACS Photonics 1, 857-862 (2014), Journal of Applied Physics 103, 103520 (2008), Physical Review B 93, 035207 (2016), Applied Physics Letters 89, 141902 (2006), J. Appl. Phys. 102, 116102 (2007), AIP Advances 10, 075025 (2020), Diamond and Related Materials 20, 849-852 (2011)].

Figure 4 | DUV laser excitation spectroscopy of photoluminescence and photocurrent at a temperature of 10 K. **a.** (top panel) Electroluminescence spectra from device A, B and C and (bottom panel) the representative photoluminescence spectrum from as-exfoliated hBN crystals at 10 K. In photoluminescence spectrum, narrow lines of intrinsic phonon-assisted emission (peaks at 5.89, 5.86, 5.79 and 5.76 eV) from the indirect gap (iX with black dashed line) are labelled as S_{1-4} lines whereas peaks at 5.63, 5.56, 5.47 eV and a shoulder at 5.5 eV in relatively broad spectral profile are labelled as $D_{1,2,4}$ and D_3 lines, respectively. **b.** 2D color map of PLE spectrum. Vertical axis, horizontal axis and the color scale correspond to the photoexcitation energy (E_x), the energy and the intensity of luminescence signal, respectively. Luminescence lines including S_{1-4} and D_{1-4} are grouped into S-series and D-series. **c.** Horizontal linecuts of **(b)** for $E_x = 6.07$ eV (top panel) and $E_x = 5.97$ eV (bottom panel). **d.** Vertical linecuts of **(b)** for S_3 -line (top panel) and D_4 -line (bottom panel). iX with black dashed line in **(c)** and **(d)** is labeled for the indirect gap of the pristine structure in hBN. **e.** Photocurrent spectrum far below and across the bandgap of hBN with the bias voltage at 3 V.

Original comment (7):

Fig r1, why would a 10 nm hBN capping layer would reduce the photocurrent to none? This is not discussed, I assume this is due to significant light absorption in the DUV range by the 10nm capping layer. But thicker capping layer on EL devices does not seem to reduce the emission, which I find odd.

Our reply:

For the question on the photocurrent, we had provided the explanation in the previous reply [reply to **Original comment (5)** in the previous reply]. Our apologies if our explanation was not clear in the previous reply. The reviewer is correct for the reason why the photocurrent is reduced almost to none. For the photoexcitation at 6.22 eV, which is well above the direct bandgap of hBN, the capping hBN layer substantially absorbs incident DUV excitation. According to the previous study [Physical Review B 13, 5560-5573 (1976)], the penetration depth is measured as few tens of nanometers, which is consistent with our observation.

On the other hand, the electroluminescence from hBN devices emerges via the phonon-assisted processes below the indirect bandgap of hBN at 5.96 eV. Thus, the capping hBN layer does not absorb the electroluminescence signal (Signals are mostly below 5.9 eV). The detailed luminescence and absorption (excitation) spectra can be found in Fig.4c and 4d, respectively. As emphasized in the abstract and the introduction of our manuscript, hBN has attracted significant interest because the radiative recombination process is surprisingly efficient due to strong electron-phonon coupling even though hBN is an indirect bandgap semiconductor [Nature Photonics **10**, 262-266 (2016), Nature Review Materials **4**, 552-567 (2019)]. Following the reviewer's comment, we have provided the discussion above in the revised manuscript in order to further elaborate the effect of the capping layer for the electroluminescence devices.

Fig. R1. I-V characteristics in the dark and under laser excitation with and without hBN encapsulation. The thickness of active hBN layers are the same (10 nm) for the device without the capping hBN and the device with the capping hBN. Without the capping layer, we observe strong photocurrent (61 nA) under the bias voltage of 1 V (blue solid line). However, the device shows only negligible photocurrent with the capping layer (red solid line). I-V characteristics of photocurrent are measured under an excitation power of 2.5 μ W at 6.22 eV. For photocurrent measurement, fourth harmonic generation of femtosecond laser pulses with 80 MHz repetition rate from Ti:Sapphire oscillator are utilized to generate the laser excitation at 6.22 eV.

Original comment (8):

3 EL intensity. I would expect to see the relationships between intensity and current, intensity and layer thickness. This is due to the charge injection efficiency and thickness of the emission layers. The intensity shall increase with the injection and injection decreases with the thickness. But in fig s2, they are all random, why is that case?

Our reply:

We appreciate the reviewer for her/his important question on the dependence of electroluminescence intensity on the emissive layer thickness. The reviewer is correct that the

tunneling injection efficiency decreases with the emissive layer thickness under a given bias voltage since the electric field decreases. However, the electroluminescence spectra in Fig.S2 are taken for the devices at the constant current at 50 μA , not at the constant voltage. The bias voltages are adjusted accordingly to match the current, which satisfies the equation below:

$$I = \frac{A_{eff}q^3mV^2}{8\pi h\Phi_B m^*d^2} \exp\left[-\frac{8\pi\sqrt{2m^*}\Phi_B^{3/2}d}{3hqV}\right]$$

V and d represent bias voltage and insulator thickness, respectively. A_{eff} is the effective tunneling area, q is the elementary charge, h is the Planck's constant, m is the free electron mass, m^* is the effective masses of electrons or holes in the insulator, and Φ_B is the tunneling barrier height for electrons or holes at the interface of the electrode and the insulator.

The fact that the devices exhibit randomly scattered external quantum efficiency (EQE) of electroluminescence means that EQE is determined by other physical parameters including the quality of hBN crystals. For example, the devices in Fig.S2 show different intensity ratio between S-series and D-series which is an indicator for the density of stacking faults in the emissive hBN layer. In the future, it will be important to characterize the role of the stacking faults on the transport and the dynamics of charge carriers in hBN, which is beyond the scope of this work.

Original comment (9):

4. Some minor points.

Experimental details and extracted results shall be shown where is handy (e.g. figure captions) for readers to put information together. I find really hard to correlate the parameters with results. For example, EL mapping in fig s6, emission in what spectral range is collected? What are the thicknesses for the hBN capping layers in fig s2? what laser is used to excite the device in fig r1, and so on?

Our reply:

We appreciate the reviewer for her/his helpful comments to improve the presentation of our manuscript. Following the reviewer's comments, we have provided the requested detailed information in the figure caption.

The spatial profiles of DUV electroluminescence at room temperature in Fig. S6 are imaged by a CCD detector with a bandpass filter with full-width-half-maximum of ~ 1 eV centered at 5.8 eV. We have provided this information in the supplementary information. The hBN capping layers in our electroluminescence devices in Fig.S2 have the thickness range from 10 nm to 50 nm. We note that, as discussed in our comment for **Original comment (7)** above, the capping hBN layers do not absorb the electroluminescence signals. For photocurrent measurement of device in Fig. R1, fourth harmonic generation of femtosecond laser pulses with 80 MHz repetition rate from Ti:Sapphire oscillator are utilized to generate the laser excitation at 6.22 eV.

3. Reply to Reviewer #3**Original comment (1):**

In my opinion the revised manuscript has been significantly improved, and the authors have responded satisfactorily to all of my comments. I therefore recommend publication of this work.

Our reply:

We sincerely appreciate the referee for her/his appreciation of the improvements in our revised manuscript and for her/his recommendation on our work for the publication in Nature Communications.

REVIEWERS' COMMENTS

Reviewer #2 (Remarks to the Author):

As per the authors' responses, they are not sure how to answer my questions. And they raise a valid point, these questions would be answered in the following project. If the editor's office is OK with the current form, I am not against publishing it.